# Genomic Characteristics of Feline Anelloviruses Isolated from Domestic Cats in Shanghai, China

**DOI:** 10.3390/vetsci10070444

**Published:** 2023-07-07

**Authors:** Jun Gao, Chengqian Liu, Jianzhong Yi, Ying Shi, Hong Li, Huili Liu

**Affiliations:** Institute of Animal Husbandry and Veterinary Science, Shanghai Academy of Agricultural Sciences, Shanghai 201106, China; gaojun@saas.sh.cn (J.G.); liuchengqian@saas.sh.cn (C.L.); yijianzhong@saas.sh.cn (J.Y.); shiyingsunny@126.com (Y.S.); lihong20061029@163.com (H.L.)

**Keywords:** viral metagenomics, feline anelloviruses, phylogenetic analysis

## Abstract

**Simple Summary:**

A viral metagenomic study of domestic cat feces revealed a high prevalence of feline anelloviruses (FcTTV) infection in domestic cats in Shanghai, China. By comparing the full-length sequences of the five Shanghai FcTTV variants, the study revealed the genomic features of the Shanghai FcTTV variants and identified recombination events between the variants.

**Abstract:**

Viral metagenomics techniques allow the high-throughput discovery of possible pathogens carried by companion animals from their feces and other excreta. In this study, the viral metagenomics of 22 groups of fecal samples from domestic cats revealed a high prevalence of feline anelloviruses (FcTTV) infection in domestic cats in Shanghai, China. Serum samples from 30 cat individuals were further detected by polymerase chain reaction, and an average positive rate of 36.67% (11/30) of FcTTV infection was found. Next, the full-length sequences of five Shanghai FcTTV variants were obtained and submitted to GenBank with access numbers OP186140 to OP186144. Phylogenetic analysis indicates that the Shanghai FcTTV variants have relatively consistent genomic characteristics, with two variants from Zhejiang 2019 and one variant from the Czech Republic 2010. The recombination event analysis of the variants showed that one variant (OP186141_SH-02) had a primary parental sequence derived from a variant (KM229764) from the Czech Republic in 2010, while the secondary parental sequence was derived from OP186140_SH-01. The results revealed that FcTTV infection is prevalent in domestic cats and that the use of viral metagenomics to rapidly identify some infecting viruses whose hosts lack clinical features would be an effective approach.

## 1. Introduction

The wide application of high-throughput sequencing technology in viral metagenomics has made it possible to detect some unknown viruses or pathogens with no obvious clinical manifestations of disease on a large scale. However, for companion animals such as dogs and cats, which have the most frequent contact with humans in cities, the viruses they contract are often overlooked or difficult to detect. The rapid detection of pathogens through viral metagenomics will help to monitor the epidemiology of viruses, examine the variation of regional epidemic strains, and assess the risk status of virus transmission in a timely manner. The viral metagenomic approach has enabled the discovery of putative novel pathogens and the identification of the etiologic agents of different diseases. It has been used in several studies including fecal samples from livestock and companion animals, providing information on the diversity of animal fecal virome and identifying potential zoonotic and emerging viruses [1].

Anelloviruses are a family of single-stranded circular DNA (ssDNA) viruses with great genetic diversity, ranging in genome size from 2 to 3.9 kb [2,3,4,5]. The “TT virus” (TTV) was first identified in the blood of a Japanese patient (initials, T. T.) with post-transfusion hepatitis of unknown etiology in 1997 [2,5]. It later became known as Torque teno virus, which is derived from the Latin words “torques” meaning “necklace” and “tenuis” meaning “thin” to describe the nature of the virus’s circular genome [4,6]. TTV have a wide range of mammalian hosts including humans, non-human primates [7,8,9,10], pigs [11], cattle [8], sheep [8], cats [7], dogs [7], rodents [12,13], bats [14], and sea lions [15], etc. However, the inability to propagate these viruses in cell culture systems has hindered a deeper understanding of such viruses.

Human TTV are thought to be associated with a number of diseases, including autoimmune diseases [16,17], hepatitis [18,19], and multiple sclerosis [20,21]. According to previous studies, human anelloviruses are subdivided into Torque teno virus (TTV, 3.6–3.9 kb in length), small Torque teno virus (TTMV, 2.8–2.9 kb in length), and medium Torque teno virus (TTMDV, 3.2 kb in length) [22]. In contrast, the feline TTV viruses (FcTTV) are much shorter, and the first FcTTV was reported by Okamoto et al. [7] in 2002. The strain was named Fc-TTV4 and has an entire genome sequence length of 2064 bases with three open reading frames (ORF1-3) encoding 436 (ORF1), 105 (ORF2), and 231 (ORF3) amino acids, respectively. While anellovirus infection is very common in mammals, only a few anellovirus genomes have been identified in the Felidae family. There is evidence of recombination within and between feline-specific anelloviruses, which supports a long coevolutionary history between the host and virus [23]. However, there is still a relative paucity of studies investigating the status of TTV carried by companion animals in close contact with humans, such as dogs and cats, and their molecular epidemiology.

Shanghai, a mega-city in eastern China with a resident population of over 24 million, has seen a steady increase in the number of pet owners over the past decade. With the global pandemic of the Corona Virus Disease 2019 (COVID-19), awareness of zoonotic diseases is increasing, and pet owners are becoming increasingly concerned about the epidemiological investigation of their companion animals. In this study, we aimed to investigate the virus infection from domestic cats in Shanghai, China. High-throughput viral metagenomics techniques were used to identify the pathogen and verify the presence of the virus infection by molecular biology methods. Through the genomic analysis of the prevalent variants, we aimed to unravel the genetic characteristics of the variants and assess whether there is a potential risk of transmission.

## 2. Materials and Methods

### 2.1. Viral Metagenomics of the Fecal Samples

In this study, 55 cat fecal samples from domestic pet cats were randomly collected from five veterinary hospitals in four districts of Shanghai, China (Minhang, Jiading, Baoshan and Jing’an). The other 55 cat fecal samples were obtained from stray cats in parks or stray animal shelters. The fecal samples from every five cats were mixed in equal amounts into one tube, with 11 tubes of samples (numbered FD1~FD11) for the pet cat group and 11 tubes of samples (numbered F1~F11) for the stray cat group. None of these cats showed any obvious signs of disease or health problems.

### 2.2. Sample Pretreatment

The fecal samples were placed in the sterile-tipped Eppendorf tube and frozen at −80 °C within 4–8 h of collection. One gram of the fecal sample was washed with five volumes of precooled sterile Stabilization Buffer, vortexed for 5 min, and after three rounds of freezing-thawing and the centrifugation of the sample at 12,000× *g* for 5 min to remove the precipitate, the cell fragments were removed by 0.45 µm and 0.22 µm ultra-filtration tube and the supernatant was transferred to the ultra-centrifuge tube containing 28% (*w*/*w* of sucrose. After centrifugation at 160,000× *g* for 2 h at 4 °C by HIMAC CP 100 wx ultra-centrifuge (Hitachi, Tokyo, Japan), and after the removal of the supernatant, the precipitate was resuspended in 200 μL SB Buffer, proportionally added into the Enzyme Mix Buffer and Enzyme Mix, and then incubated at 37 °C for 60 min; Stop Solution (2 μL) was added in proportion, mixed well, responsible for inactivating the enzyme reaction at 65~75 °C for 10 min, centrifuged at 2000× *g* for 5 min, and able to store 200 μL of the supernatant at −20 °C for subsequent experiments. The SB buffer, enzyme mix buffer, and enzyme mix are derived from the VirOne Kit (Guangdong Magigene Biotechnology Co., Ltd., Guangzhou City, China). The co-extraction of the DNA and RNA viruses in pretreatment samples was completed with Magen kits (R6662-02 MagPure Viral DNA/RNA Mini LQ Kit) (Magen Biotechnology Co., Ltd., Guangzhou, China) and the whole genome was amplified with the Qiagen kit (150054 REPLI-g Cell WGA and WTA Kit) (QIAGEN Group, Venlo, The Netherlands) and finally amplified using Thermo NanoDrop One, Life Technologies Qubit 4.0 (Thermo Fisher Scientific, Waltham, MA, USA), and 1.5% of agarose electrophoresis for the quality control of the products.

### 2.3. Library Generation and Sequencing

Sequencing libraries were generated using the NEB Next^®^ Ultra™ DNA Library Prep Kit for Illumina^®^ (NewEngland Biolabs, Ipswich, MA, USA) following the manufacturer’s recommendations. The main steps include: (1) DNA sequence amplification and product fragmentation; (2) end repair and 3′ end plus A; (3) adaptor ligation, fragment selection, and purification; and (4) PCR amplification and purification. The library quality was assessed using the Qubit R dsDNA HS Assay Kit (Life Technologies, Grand Island, NY, USA) and Agilent 4200 system (Agilent, Santa Clara, CA, USA). High-throughput sequencing was conducted on an Illumina Novaseq 6000 and 150 bp paired-end reads were generated by the Magigene Company (Guangzhou, China).

### 2.4. Data Analysis

Raw Data processing using Trimmomatic (Bolger et al., 2014) [24] (v0.36) was conducted to acquire Clean Data for subsequent analysis. The sliding-window algorithm was used to trim the reads with low quality after removing the adapters using Cutadapt (v1.2.1) (Technical University Dortmund, Dortmund, DE, Germany). All clean reads were mapped to the host reference genome (GCF_018350175.1) of *Felis catus* (domestic cat) and the ribosomal database (silva.132) utilizing BWA software (Li and Durbin, 2009) [25] (v0.7.17). Clean reads without host sequences and ribosomes were de novo assembled for each sample and mapped to the GenBank non-redundant nucleotide (NT) database to identify virus reads primarily. The obtained contigs are compared with the virus database, and those with a mapping length of ≥100 bp and e ≤ 1 × 10^−5^ are used as the criteria to determine the detection of the virus.

### 2.5. Validation of FcTTV Infection

Many of the viruses detected by viral metagenomics may be related to the food ingested by the cats or the environment in which they live. The purpose of this study is to understand real virus infection in cats, and identify which viruses have high infection in pet or stray cats. Thus, following viral metagenomics analysis, another group of 30 serum samples from cats of different ages and different health conditions, collected from veterinary clinics in Minhang District, Shanghai, China, from June to October 2021, were used for a validation study of FcTTV infection.

Total viral DNA was extracted from the above 30 serum samples using a viral DNA extraction kit (Takara Biomedical Technology (Beijing) Co., Ltd., Beijing City, China) according to the manufacturer’s instructions. The extracted DNA was used for the detection of FcTTV by nested PCR. The primer pairs (Table 1) of FcTTV-P1 and FcTTV-P2 was used for the first round of amplification, and FcTTV-P3 and FcTTV-P4 were used for the second round of amplification [26]. Both sets of primers were localized in the highly conserved untranslated region (UTR) of the TT virus genome. The PCR positive samples were presented with a band at 314 base pairs.

The first round of the PCR reaction system included 2.0 μL of template DNA, 1.0 μL each of primer (10 μmol/L), 12.5 μL of 2 × Phanta Max Master mix (Vazyme Biotech Co., Ltd., Nanjing City, China), and sufficient ddH_2_O to increase the volume to 25 μL. A 2 μL aliquot from the first round of PCR was used in the second reaction. The first and second round of PCR with both sets of primers were performed under the same conditions. Amplification was carried out as follows: one cycle at 95 °C for 5 min, followed by 35 cycles at 95 °C for 15 s, primer annealing at 57 °C for 15 s, extension at 72 °C for 40 s, and final extension at 72 °C for 7 min. PCR products were detected by electrophoresis in 1.2% of agarose gel using 6 μL of aliquots in 1.2% of agarose gels in 1 × TAE.

### 2.6. Full-Length Sequence of the Shanghai FcTTV Isolates

According to the FcTTV genome sequence (accession number HM142588) registered in GenBank, four primer pairs (Table 1) were designed to amplify the full-length sequences, which were obtained through nested PCR amplification, Sanger sequencing, and sequence assembly. Primers FcTTV-P1Q and FcTTV-P2Q were used for the first round of amplification and FcTTV-P3Q and FcTTV-P4Q were used for the second round of amplification.

Another sequence was amplified using primers FcTTV-P5Q and FcTTV-P6Q for the first round of amplification and FcTTV-P1 and FcTTV-P2 for the second round of amplification. Amplification was completed using Phanta Max Master mix (Vazyme Biotech Co., Ltd.).

The PCR positive samples were presented with a band at 1528 and 579 base pairs. PCR products were purified by the MinElute PCR Purification Kit 28004 (Qiagen, Hilden, Germany), ligated into the pMD20-T Vector (Takara Biomedical Technology (Beijing) Co., Ltd.), and cloned according to the manufacturer’s instructions. The clone products were sent to Sangon Biotech (Shanghai) Co., Ltd., Shanghai City, China, for sequencing.

### 2.7. Phylogenetic Analysis of the FcTTV Shanghai Isolates

The five Shanghai FcTTV isolated sequences obtained in this study were aligned along with those of the 75 TTV related strains retrieved from GenBank, such as Human TTV virus, Porcine TTV, Felis TTV, Canine TTV sequences, Lynx rufus, and Puma concolor TTV sequences (Appendix A). The phylogenetic tree was inferred based on the open reading frame 1 (ORF1) amino acid sequences of the TTV virus strains using the Neighbor-joining (NJ, 500 bootstrap replicates) and maximum likelihood (ML, 100 bootstrap replicates) approaches implemented in MEGA X (Mega Software, Auckland, New Zealand, accessed on 3 May 2023) [27] and visualized in iTOL (https://itol.embl.de/itol.cgi, accessed on 5 May 2023) [28]. The LG + G + F model obtained the lowest Bayesian information criterion (BIC) scores and was set as the best amino acid substitution model conducted in MEGA X in the ML method. The phylogenetic tree branches corresponding to partitions reproduced in less than 50% bootstrap replicates are collapsed. Recombination and genome component reassortment are processes that strongly impact the evolution of many virus species. Therefore, recombination events were also detected using RDP4.101 [29] for these 80 TTV sequences.

## 3. Results

### 3.1. Cat-Associated Viruses Detected by Viral Metagenomics

A variety of viruses, including plant viruses, animal viruses, and even some viruses from humans, were identified in cat fecal samples through viral metagenomics studies. A total of 493 viruses and 657 viruses were detected in the domestic pet cat group (FD group, FD1-FD11) and the stray cat group (F group: F1–F11), respectively (Appendix A). It is worth noting that several feline viruses had high detection rates in both groups of samples, especially the feline anellovirus (FcTTV, Figure 1). The reads obtained by sequencing and the assembled contig statistical data are available in Appendix A.

### 3.2. Validation of FcTTV Infection

The PCR experiments yielded a product band consistent with the expected length of 314 bp, which was confirmed by sanger sequencing and blast comparison. The results confirmed the presence of highly positive FcTTV infection in domestic cats. Eleven out of thirty randomly collected serum samples were positive for FcTTV, indicating that the mean positive rate of FcTTV obtained in this study was 36.67% (11/30) in Shanghai domestic cats, which confirmed that the virus was not due to food ingestion or environmental sampling. The PCR product electrophoresis figure was shown in the Appendix A.

### 3.3. Full-Length Sequence of the Shanghai FcTTV Isolates

Five full-length sequences of FcTTV Shanghai variants were obtained from the fecal and serum samples through PCR and sequence assembly. We submitted these five sequences, FcTTV_SH-01~05, to GenBank (https://www.ncbi.nlm.nih.gov/, accessed on 4 August 2022) and obtained the accession numbers OP186140~OP186144, respectively. The full length of the genome of these five variants ranged from 2054 to 2058 bp, and the pairwise distance between the five sequences ranged from 0.0108 to 0.0926. The average nucleotide composition of them is 21.8% T, 25.8% C, 25.6% A, and 26.8% G, respectively.

### 3.4. The Results of Phylogenetic Analysis

Phylogenetic tree analysis of the 80 ORF1 sequences based on the maximum likelihood method showed that the five Shanghai FcTTV isolates in this study were in the same large branch as another domestic cat source of FcTTV (Figure 2). The results also revealed that TTV viruses of different species demonstrated strong species specificity and length differences. For example, the average length of ORF1 of human TTV viruses was more than 2K, while the ORF1 of feline species was relatively much shorter, and the ORF1 of all five strains isolated in this study was around 1.3K, and two FcTTV2 variants were even shorter at 1182 bp (JF304938 VS430008, USA 2011) and 1215 bp (EF538877 PRA1, France 2007).

The largest ORF in the FcTTV genome is ORF1, which encodes a total of 436 amino acids, all starting at position 438 of the genome, but some strains have base deletions in ORF1, and three (SH-01/03/05) of the five Shanghai isolates in this study were missing CGA bases at positions 131-133 on ORF1, which resulted in an arginine deletion. All five Shanghai isolates were missing six bases at 817-820, 827, and 828, which also resulted in corresponding amino acid changes (Figure 3). These missing features differed from the two Shanghai isolates (HM142589_SH-F1 and HM142589_SH-F2) reported in 2011 [30], and indicate that the five FcTTV Shanghai isolates are closely related to two sequences (MT078975_GX26_2019 and MT078974_GX27_2019) isolated from Zhejiang, China and one sequence (KM229766_278_2010) reported from the Czech Republic.

The phylogenetic consensus tree involved 80 complete ORF1 amino acid sequences of TTV virus inferred by the Neighbor Joining method (NJ) and Maximum Likelihood (ML) method conducted in MEGA X [27] and visualized in iTOL [28]. The bootstrap confidence interval [31] is shown as a dot size on the branch, and the corresponding confidence value range is in the upper right corner of the legend. The five Shanghai FcTTV variants (dark red) are clustered in the blue FCTTV branch of domestic cat origin. The pink (human TTV), orange (porcine TTV), and green (canine TTV) branches showed a closer phylogenetic relationship. The TTV sequences in grey block were mainly isolated from wild Lynx rufus and Puma concolor.

Three of the five Shanghai FcTTV variants (SH-01/03/05) in this study were missing “GAC” bases at positions 131-133 in ORF1, which resulted in an arginine deletion. All five Shanghai isolates were missing six bases at the 817-820, 827, and 828 positions in ORF1. These deletion features were also present in two Chinese variants from Zhejiang (MT078975_GX26 and MT078974_GX27, 2019) and one variant from the Czech Republic (KM229766_278, 2010).

### 3.5. The Recombination Event Detection

The RDP detection of 80 ORF1 sequences revealed that there was a recombination event in the sequence OP186141_SH-02, whose major parental sequence originated from KM229764_183_Czech Republic 2010, while minor parental sequence originated from OP186140_SH-01, and all six default detection methods (RDP, GENECONV, Bootscan, Maxchi, SiSsc, and 3Seq) in the RDP program had significant p values (i.e., *p* value of RDP = 1.343 × 10^−23^, Appendix A).

This also echoes the sequence deletion feature described above. The OP186141_SH-02 sequence does not have a “GAC” deletion at positions 131-133, which is consistent with KM229764_183_Czech Republic 2010, while it has the same six-base deletion as SH-01 at 817-828, as the recombination breakpoint position occurs at the 664 bp site in ORF1 of FcTTV (Figure 4).

The recombination event was detected in the sequence OP186141_SH-02, whose major parental sequence originated from KM229764_183_Czech Republic 2010 and minor parental sequence originated from OP186140_SH-01 with the significant p value of RDP = 1.343 × 10^−23^.

## 4. Discussion

TTV variants in non-human primates are species-specific, and the genomic organization and proposed transcriptional profile of TTVs infecting other mammalian species are similar to those of human TTVs, so it has been proposed that there is the co-evolution of TTVs with their hosts [32]. A study conducted in the Czech Republic reported that a total of 37 out of 110 cats were identified with FcTTV infection by the nested PCR technique (33.63% positivity rate) [26]. This is very close to the mean positive rate of 36.67% for FcTTV in Shanghai domestic cats obtained in this study. Therefore, it was shown that FcTTV infection was prevalent in domestic cats. Moreover, the genomic characteristics of the five variants obtained in this study differed from the two Shanghai variants reported in 2011, while they were more closely related to the 2019 Zhejiang variants and another earlier Czech variant, suggesting the presence of the inter-regional transmission of FcTTV.

In the phylogenetic analysis of 80 sequences, two variants (EF538877 PRA1 France 2007 and JF304938 VS430008, USA 2011) were considered as FcTTV2 variants. The PRA1 variant was reported in 2007 with another variant (EF538878 PRA4 France 2007) identified from cat saliva, and the ORF1 homology between these two variants was only 54%, showing great variability [33]. These two variants were also included in our study and the phylogenetic tree showed that the PRA1 was closer to the FcTTV derived from Lynx rufus and Puma concolor, while PRA4 was homologous to the other domestic cat sources of FcTTV. The five FcTTV Shanghai variants obtained in this study are all clustered together with the PR4 in this branch from domestic cat. Since FcTTV2 (PRA1) is apparently a different strain from the branch in domestic cats, is it mainly found in large wild Felidae animals? How it infects domestic cats, or whether the unique variant is detected due to the exposure of domestic cats to secretions from other wild Felidae animals, still has not been fully elucidated.

Although TTV are clearly species-specific [9], it is noteworthy that our results confirmed a high confidence level of recombination events occurring in the Shanghai FcTTV variants. Previous studies have identified TTV sequences in 19% of chickens, 20% of pigs, 25% of cattle, and 30% of sheep, speculating that domesticated farm animals have the potential to become a source of human TTV infection [8], and likewise as companion animals in close contact with humans; thus, it is reasonable to be concerned about the TTV mutations they carry, as this could also be an important source of the genetic recombination evolution of human TTV in the future. Clear evidence of recombination has been described in human and non-human TTV [34,35]. Species richness and diversity were close to saturation in each anellovirus genus, and recombination was found to be the main factor promoting diversity [36].

A study from the Republic of Korea reported a 9.6% positivity rate for canine TTV virus (Torque teno canis virus, TTCaV) in 135 fecal samples from dogs, and phylogenetic analysis of three full-length sequences also monitored a recombination event between the strains isolated from the Republic of Korea and China [37]. Sun et al. (2017) [38] investigated the prevalence of TTCaV in Guangxi province, China and found a positive rate in domestic dogs at 7% and the presence of co-infection with canine parvovirus (CPV). Zhu et al. [30] collected 26 cat samples (16 serum and 10 stools) from a vet clinic in Shanghai in March 2010 for FcTTV investigation. The results showed that 2 out of 16 serum samples had a positive rate of FcTTV detected by the PCR test (12.5% positive rate). Of course, due to the limitation of the number of samples tested and the difference in the number of sampling sites, we cannot be sure whether the positive rate of FcTTV in Shanghai cats has increased after more than 10 years, but it is certain that a higher rate of FcTTV positivity does exist in Shanghai domestic cats. However, FcTTV epidemiological investigations in domestic cats are still not adequately studied. Currently, the virus cannot be cultured in vitro, so the pathogenicity, virus transmission, and evolutionary mechanisms of FcTTV should be further elucidated. 

In this study, FcTTV was detected by the viral metagenomic method not only in fecal samples, but also in the serum of the cat by the PCR method, thus proving that the cat was indeed infected with the virus. However, most of the viruses found in the metagenomic study were more likely to be related to the feeding and living environment of the individuals. Therefore, the greatest advantage of viral metagenomics is that they allow the high-throughput detection of pathogens that animals may carry, and then validate the infection with the relevant target pathogens by other means. On the other hand, whether viral metagenomics can better facilitate the detection of some zoonotic pathogens in companion or domesticated animals is a question that deserves more in-depth study and exploration. For example, some studies have reported the detection of SARS-CoV-2 variants in the feces or nasal swabs of domesticated dogs and cats that have been exposed to owners who tested positive for SARS-CoV-2 [39,40].

## 5. Conclusions

This study reveals the promising potential of using high-throughput viral metagenomics to discover pathogens, especially for some pathogens that are clinically asymptomatic or more difficult to isolate. The genomic characteristics of the five Shanghai FcTTV variants illustrate that the variants prevalent in the Shanghai region share relatively consistent genomic features with two FcTTV variants from Zhejiang in 2019 and an earlier variant from the Czech Republic in 2010. The occurrence of recombinant events between the FcTTV strains emphasizes the importance of the epidemiological surveillance of endemic strains. Due to the limited number of samples tested in this study, it is not sufficient to comprehensively assess the epidemiology of FcTTV in domestic cats in Shanghai, and more comprehensive data accumulation is needed in the future.

## Figures and Tables

**Figure 1 vetsci-10-00444-f001:**
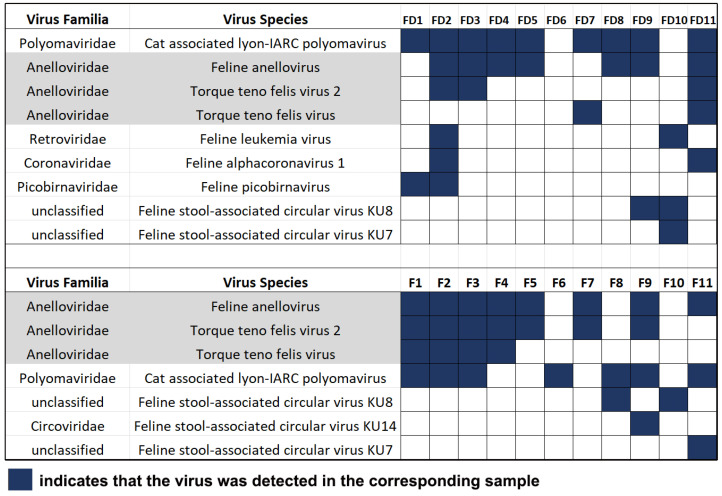
Viral metagenomics indicates that feline anellovirus (FcTTV) has a high detection rate in domestic cat fecal samples. The gray ones all indicate FcTTV.

**Figure 2 vetsci-10-00444-f002:**
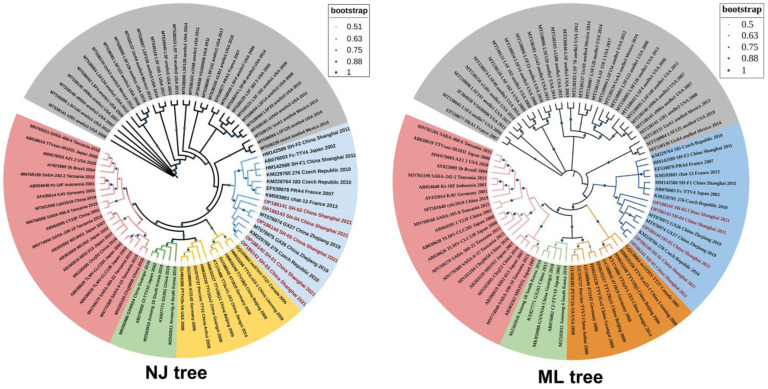
Evolutionary analysis of FcTTV by phylogenetic tree.

**Figure 3 vetsci-10-00444-f003:**
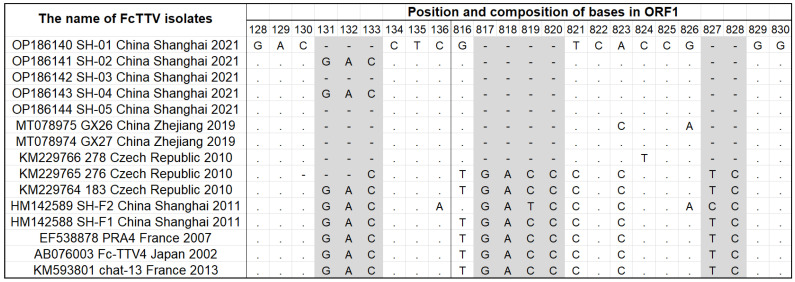
The sequence genomic characterization of the 15 FcTTV variants.

**Figure 4 vetsci-10-00444-f004:**
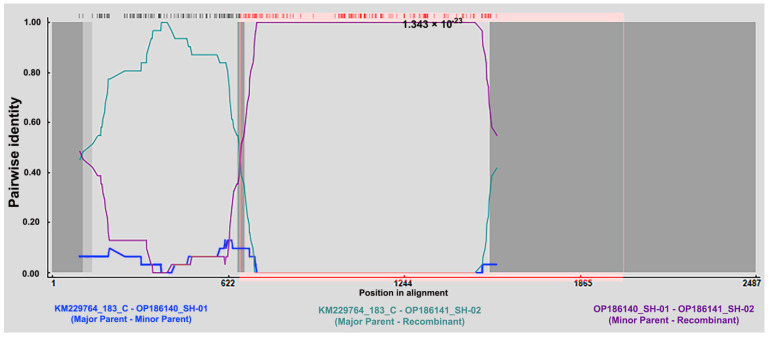
Recombination events were detected in the ORF1 of OP186141 SH-02 variant.

**Table 1 vetsci-10-00444-t001:** The PCR primers were used to amplify the FcTTV sequences.

Primer Name	Sequence (5′-3′)	Product Size (bp)	Position (nt) ^a^
FcTTV-P1	GTAAGTACACTGACGAATGGCT		8-585
FcTTV-P2	CAGTTACCACAGTTCGAGGTCGT		
FcTTV-P3	ACTGGTGACAGGACGTGCGA	314	50-363
FcTTV-P4	TGCGGAGACAAGTTGCTTCC		
FcTTV-P1Q	TCGCACGTCCTGTCACCAGT		344-69
FcTTV-P2Q	GGAAGCAACTTGTCTCCGCA		
FcTTV-P3Q	TCAGCCATTCGTCAGTGTACTTACT	1528	567-31
FcTTV-P4Q	CGACCTCGAACTGTGGTAACTG		
FcTTV-P5Q	GTTAGTGTTGCTTTACGGCAG		1969-753
FcTTV-P6Q	TGAACCAGTGGTGTCCCCAG		
FcTTV-P1	GTAAGTACACTGACGAATGGCT	579	7-585
FcTTV-P2	CAGTTACCACAGTTCGAGGTCGT		

^a^ Oligonucleotide positions are referred to the FcTTV isolate SH-F1, complete genome (GenBank accession number: HM142588).

## Data Availability

Viral metagenomics results are available in Appendix A.

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
