# Peer review of "Genomic Characteristics of Feline Anelloviruses Isolated from Domestic Cats in Shanghai, China"

_vetsci, 2023, doi:10.3390/vetsci10070444_

Round 1

Reviewer 1 Report

Gao et al. present a feline viral metagenomics study with with owned and stray cats. Initial fecal results were further investigated with another study with a different set of cats. Overall, this is a solid study with some flaws that could be addressed with revisions to the text to improve clarity and reproducibility. The sample selection for the second study on serum is not well described, and the results do not clearly annotate the population and sample type from which they were derived. The discussion is also overly brief. Some additional specific comments are below:

Line 50: references are needed here

Section 3.1 What proportion of the genome was recovered in each pool?

Section 3.2: no figure or table is associated with this section

Line 183 and Fig. 3: Specify what type of samples the complete genomes came from (feces or serum) and if they were from the strays or the owned cats.

Fig. 4: The text in yellow is illegible.

Line 256: Are the authors implying here that they found such a difference?

Section 3.5: From what host species and sample types did these strains originate? Line 274 discusses cross-host recombination, but I don't see any evidence of this.

Discussion: Only one sentence is devoted to the other viral species detected. This is a missed opportunity to highlight those results and compare to other studies. For example, it is surprising that more of the owned cats had feline coronavirus than the stray cats.

There are grammatical errors throughout the text.

Author Response

We thank the reviewers for their valuable comments. We have revised the manuscript, and hopefully to have answered all the questions, please see the attachment.

Reviewer 2 Report

I read with interest this paper investigating Anelloviruses in domestic cats by utilising high throughput sequencing methodologies for initial detection and confirmatory PCR. Although not ground breaking research, the data presented aids in further understanding these recently discovered viruses, by including country specific data from China and attempting to situate within the context of published sequences.

In general the paper was adequately written, but found myself wanting more detailed  analysis on the number of positive cats (by PCR) across the faecal sample groups and more information on how unique the recombination event is within this group of viruses. The discussion could have focussed more on the novel findings of this work and also on the other viruses identified in the metagenomics analysis. The conclusions regarding pet movement need to be validated by data or removed.

Below are a number of specific comments/questions that the authors need to address:

1.      In M&M 2.2 line 83-87, please specify what the SB buffer, enzyme mix buffer and enzyme mix are.

2.      In M&M 2.2 line 87-90, even if you followed the manufacturer protocol, please briefly describe how the whole genome was amplified, and how genome was generated (by DNA quantification for example).

3.      In M&M 2.3 even if you followed the manufacturer protocol, please briefly describe how the library prep was completed, and if there was any cut off applied for the quantity used.

4.      Table 1 please add genome positions for each primer

5.      Results 3.1: in M&M you state that sequences under 100bp were excluded, but how long were the reads that you have included? It would be helpful to provide as a minimum the range with a mean length. Numbers of contigs obtained, and % coverage across the genome would also be interesting to the reader. The supplementary data is hard to interpret as its raw data, so a supp or additional table with this information would be helpful.

6.      3.2 validation, I wonder why you didn’t attempt to collect serum from the same individuals that you got the stool samples from?

7.      3.3 full length sequencing. Please state in total how many positive samples you had (from faecal samples from domestic cats, faecal samples from stray cats and serum samples from domestic cats), so its clear to the reader. Is there a reason why only 5 full length genomes were obtained? Could you have used PCR to amplify ORF1 from more positive samples to enhance the phylogenetic analysis?

8.      Discussion: You start by stating that TTV are species specific, yet go on to talk about recombination across species, if this is a risk factor please explain how and if its been observed before.

9.      Discussion: You state that most of the viruses detected were likely related to the feeding and living environment of the individuals. Do you have any experimental evidence for this? If so please add this evidence in the results. For example you detect Feline leukaemia virus in 2 pools of your domestic animal faeces, why would this not be evidence of a real infection?

10.  Line 258-270: this paragraph is hard to read/interpret the relevance.

11.  Lines 271-280: is your finding of recombination the first for TTV? If so this is a major finding, if not please introduce other recombination events that have been described and discuss how your findings fit with them.

12.  Conclusions: line300-301: where is the evidence for pet trade involvement? Is there geographic groupings for these viruses?? If so this needs to be introduced and discussed before you make this conclusion.

The abstract and discussion could benefit from some editing to improve the English, otherwise the English is OK.

Author Response

We thank the reviewers for their valuable comments. We have revised the manuscript and hopefully have answered all the questions, please see the attachment.
